# Complexity and Timeliness of the Term "Christendom" for Ecumenical Ecclesiology

Filip Krauze 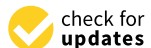

Faculty of Theology, John Paul II Catholic University of Lublin, 20-950 Lublin, Poland; filip.krauze@kul.pl

**Abstract:** The purpose of this paper is to examine whether and why the term "Christendom", despite its ambiguous historical connotations, can be taken into account in contemporary ecumenical ecclesiology. This will be performed through a linguistic, historical, and theological analysis of the term in question. Its uses in the literature and occurrences in the historical contexts have been reviewed. Particularly important in this case turned out to be St. Augustine's work "The City of God", excerpts of which shed light on some stereotypes that can place the term "Christendom" in merely political meaning. For correct discourse, one needs, on the one hand, an awareness of the traumas that the community of believers have gone through in the history of humanity along with the entire humanity, and on the other hand, the outright revolution that has taken place in post-conciliar theology. It seems that the term "Christendom" retains its relevance especially in the context of the conciliar images of the Kingdom of God and the theology of the Church of Christ. In another way then, "Christianity" reminds one of the Church's rootedness in a particular place, time, and culture, providing a tool for the humble contextualization of ecclesiology in the history of humanity.

**Keywords:** Christendom; Christianity; Church of Christ; City of God; Kingdom of God

## 1. Ecumenical Ecclesiology

In her reflection on the changing relationship between Catholic ecclesiology and ecumenism, Kristin Colberg expressed the following conclusion: "Today the church is adapting to a new context, one that demands time for exploration and discernment. Both ecumenism and ecclesiology are engaging in the creative work of self-transformation in order to meet the needs of the new era. In their transformations, we see that the close relationship between them not only endures but is a key for coming to a deeper understanding of the Christian identity and speaking meaningfully in the current environment." (Colberg 2018). This hope expressed regarding the synodal revival in the Catholic Church inspires further exploration, both in the layer of dialogue and in the layer of the common tradition of the Church and Churches. In this context, it is worth noting some attempts to define such a common ecclesiology. These can include the collective work "*Ecumenical Ecclesiology, Unity, Diversity and Otherness in a Fragmented World*", published in the series "Ecclesiological Investigations" (Thiessen 2009), and the works of the Lublin School of Fundamental Theology, represented by Cardinal Stanislaw Nagy (Napiórkowski 2023).

One of the challenges of contemporary ecumenical ecclesiology was pointed out by Brian P. Flanagan (2009) in the already-cited volume edited by Ges Elsbeth Thiessen, in context of the negotiation of inner-denominational otherness: "One indication of the complexity of these issues is the lack of good vocabulary to talk about otherness between and within those churches".[1] As frankly noted by Simon Hansbauer, a participant of ecumenical dialog conducted in the international environment, "A full week of speaking mostly English, it also meant, for me, sometimes changing between German, English and French within a couple of minutes. Some terms like "communion" or "consecration" have different aspects of meaning in the English and German languages, so "Communion of

Churches" (the title of the course) also meant the need to find a common vocabulary." (Hansbauer 2023).

Thus, the complexity of the problem raised is influence by many factors: lexical differences in different languages, the diversity of ecclesiastical traditions, and the baggage of historical experience that has shaped them. It seems to be no different with the terms "Christianity" and "Christendom", which in this study will serve as an illustration of the problem outlined and will lead, at least to some extent, towards its resolution.

## 2. Christendom—Meaning and Language

"Christendom" (ger. Das Christentum/Die Christenheit)[2] is being defined as "the worldwide body or society of Christians; the Christian world. The word is recorded from Old English (in form crīstendōm), and comes from crīsten 'Christian' + -dōm 'domain'".[3] The Merriam-Webster Online Dictionary defines it in a straightforward way: in the first sense, "Christendom" equals "Christianity", and in the second, it is "the part of the World in which Christianity prevails".[4] According to Britannica's article, recently revised by Adam Augustyn, "After the dissolution of the Roman Empire, the idea arose of Europe as one large church-state, called Christendom. Christendom was thought to consist of two distinct groups of functionaries: the sacerdotium, or ecclesiastical hierarchy, and the imperium, or secular leaders".[5] Various treatises having specifically "Christendom" in their title can be found either in German or English literature (Hilbert 1908; Haecker 1946; Bloch 1968; O'meara 1961; Tyerman 2005; Jenkins 2002; Holland 2009; Heather 2023). Although some of the works mentioned may treat "Christendom" interchangeably with "Christianity", from their very subject matter, a certain picture emerges of the distinction between the belief layer of the religion under discussion and its social influence, and between religious structures and their cultural impact, what Marian Rusecki called the historiotransformative argument of Christianity's credibility (Krauze 2023b, p. 40). Significantly, the entry "Christendom" is not offered by The New Catholic Encyclopedia,[6] while there are references to "Western" and "Orthodox" Christendom in the volume, which positively reflects the historical-cultural and ecumenical sensibilities of the researchers, as well as contexts indicating the use of the term in question in the sense of "the Christian World".[7]

In addition to hermeneutical problems, it is also necessary to point out the linguistic problem in properly conveying the content of the term "Christendom". There are languages for which the specificity in question remains almost invisible. For example, in Polish, "Christendom" translates simply as "Christianity", and there is no separate term enabled to express the thought of Douglas Hall and other authors addressing the issue.[8] Such a phenomenon opens up a space for ambiguity and can generate certain legitimate, founded as they are in historical experience,[9] concerns and true, though counterproductive, fears: here, depriving Christianity of "dominion or sovereignty" will simultaneously mean the end of Christian faith and morality, at least in the public square.[10]

The following examples can serve as an illustration of the plurality of approaches to the term "Christendom" in the mentioned historiotransformative sense. One of the precursors of German Romanticism, Novalis (Georg Philipp Friedrich Freiherr von Hardenberg), whose 1799 essay was posthumously titled "Die Christenheit oder Europa. Ein Fragment", can also be considered one of the greatest supporters of the "Golden Age of the Holy Middle Ages". Unlike Emmanuel Kant and Johan Gottlieb Fichte, and like Friedrich Schlegel, but in a more mature and developed manner, he advocated a new understanding of cosmopolitanism, by no means based on self-interest and social antagonism. He polemicized with representatives of the Enlightenment, spinning a eutopia of a peaceful future for Europe based on the Christian order known before the Reformation (Kleingeld 2008). Thus, as a Protestant, in his last years, he was turning to Catholicism (Haussmann 1912, 1913). On the other hand, Douglas John Hall, the renowned theologian of the United Church of Canada pondering the future of Christianity in his work "The end of Christendom and the Future of Christianity" in passing, as it were, gives some definition of "Christendom": "That coveted future is what I mean when I use the term "Christendom"—which means literally the

dominion or sovereignty of the Christian religion. Today Christendom, so understood, is in its death throes, and the question we all have to ask ourselves is whether we can get over regarding this as a catastrophe and begin to experience it as a doorway—albeit a narrow one—into a future that is more in keeping with what our Lord first had in mind when he called disciples to accompany him on his mission to redeem the world through love, not power" (Hall Douglas 1997, p. IX).

Chronologically between the above exemplary approaches to the phenomenon of "Christendom", which are, after all, quite distant in time, a debate on Church and politics, spanning from the 1930s till the 1950s is noteworthy. These voices have been laboriously collected recently by Gianmaria Zamagni in a work entitled "The end of the Constantinian age" and the models from history for a "new Christendom". A religious-historical investigation." (Gianmaria Zamagni 2018, 2011). This opus, by design, refers to both the thought of Marie-Dominique Chenu and the unfinished studies of Giuseppe Alberigo (died 2007), undertaken on the occasion of the 1700th anniversary of the Edict of Milan. In addition to the aforementioned Chenu, such thinkers and their contexts are being featured, such as Miguel Unamuno, Friedrich Heer, Etienne Gilson, Emmanuel Mounier, Jacques Maritain, Ernesto Buonaiuti, Nikolai Berdjaev, and others. Their common denominator is the realization, expressed in Cardinal Suchard's milestone pastoral letter (Suhard 1947), that in some sense they are living in a time of breakthrough, leading to a new reinterpretation of the role of Christianity. According to Zamani, Suchard advocated an incarnationalist model of the Church's presence in the world, "never binding itself to external forms, but adopting them only for the purpose of sanctification".[11] Today, in the area of socio-political doctrine, Christianity no longer clashes as clearly as it did in the early middle of the last century with modernist totalitarianisms. One of the significant problems of the late 20th century is the relationship of the Christian social tradition to postmodernism, as reflected, for example, in Karl Gabriel's research (Gabriel 1992), which Christian Spieß summarized as follows: "In the context of a reflexive and radicalized modernization, church-based Christendom is also experiencing increased pluralization and polarization without, however, dissolving. The study offers a comprehensive historio-empirical and conceptual-theoretical reflection and classification of the situation of Christianity, especially Catholicism, in the modernization process. On a more general level, it offers a critical approach to the secularization theorem and to the question of the role of religion in modernity." (Spieß 2019).

### 3. Christendom—Term's Witness to (R)Evolutions

"Dominion or sovereignty" could undoubtedly support the visible growth of Christianity: psychologically, it is easier to belong to the majority and receive broad social acceptance of one's beliefs than to be in the minority punished for every manifestation of otherness. However, the new faith required outright non-conformism. Biblical sources point to miracles and unusual signs as the attractive force of the new religion.[12] The apologist Tertullian, on the other hand, explains the phenomenon of the growth of Christians through the moral power of the testimony of martyrs, which tradition encapsulates in the maxim "Sanguis Martyrum Semen Christianorum". In doing so, he gives many examples of the moral strength of non-Christians who, for a just cause, suffered various types of torture and gave their lives, which paradoxically adds to the rationality of the Christian testimony that it is not just about some incomprehensible suicidal instinct.[13] On the contrary, "Being in the majority" can also carry the seeds of paganism. As Joseph Ratzinger notes: "the three main forms of polytheism are the worship of bread, the worship of love and the idolization of power. All three paths are aberrations; they make absolutes out of what is not in itself the absolute and thereby makes slaves of men. (...) As a declaration of war on this threefold worship this confession of faith (...) is a renunciation (...) of gods of one's own or, in other words, of the deification of one's own possessions, something which is fundamental to polytheism." (Ratzinger 1970, p. 74) In the Polish edition, "deification of one's own possessions" was translated as "idolization of the peculiar". Being "salt of the earth" excludes Christianity's "peculiarity". Probably in this sense, Jesus Christ, who, after

all, offers "peace that the world cannot give",[14] at the same time also teaches about the fire, sword, and division that he came to cast upon the earth,[15] which is why there have been times in history when revolutionaries with beliefs far removed from the Gospel have also referred to Jesus.[16]

The rapid growth of Christians in the pre-Constantinian period cannot be explained by the opportunism of the masses nor being easily exposed in sociological terms. As Kenneth Latourette stated, "One of the most amazing and significant facts of history is that within five centuries of its birth Christianity won the professed allegiance of the overwhelming majority of the population of the Roman Empire and even the support of the Roman state." (Latourette 1964, p. 65). The dynamic growth of the number of Christians in the first centuries makes for a history of its kind, so much so that its reliable description calls for various and complex methods that can shed new light on the phenomenon, including numerical analysis and computer modeling.[17] According to Willem Akkerhuys Dreyer, "Quantitative modelling has serious shortcomings and cannot present us with definite answers on the growth of the early church, due to many variables (...) However, quantitative modelling can help to investigate the factors that scholars have suggested drove conversion. The majority of studies of early Christianity appropriately focus on textual interpretation. By using textual sources in combination with sociological and quantitative modelling, new lines of enquiry open up." (Dreyer 2012).

The shape of the presence of the Christian religion in the world (Christendom) evolved and revolved not only quantitatively, and in more ways than the clash of faith-paganism alternative. Chronologically, the earliest and, as we see in Zamagni's already cited synthesis, recurrent (if only in the 20th century) revolution is the prospect of an "apocalypse in history", that is, the end of the world as we know it,[18] alongside its embodied shape of religion. One is not dissuaded from such reflections by the climate of the 2020s, a world, so to speak, helpless in the face of the prospect of the outbreak of another world war hanging in the balance, as well as multiple climate-related, biological, technological, and social threats.[19] The fifth century brought the fall of the Roman Empire; the eleventh—fruition of the schism between the Christian East and West; the sixteenth—the war within Western Christendom; and the nineteenth—the collapse of the Church State.[20] It is no coincidence that when Christians, including Catholics, came to grapple with the trauma of World War II, theologians sensitive to the consequences of the revolutionary battering were put to work as the fathers of Vatican II; the aforementioned Marie-Dominique Chenu and Yves Congar, who at the dawn of World War II wrote about the painful crisis of Christian civilization as follows: "The gravity of the situation lies chiefly in the fact that we find ourselves faced not only with social schisms which rend, so to say, the outward texture of the Church as a society, but with schisms in the faith itself, the Christian treasure which we hold in common. Some would keep this item, some that, of the indivisible heritage of the Crucified, whose seamless garment was spared even by the soldiers who nailed Him to the cross: all wanting to keep what suits their own temperament, having lost the sense of their incorporation in the whole. We have no right to look upon this dismembering of Christendom as permanent." (Congar and Christendom 1939) The past and present condition of Christendom cannot then be fully understood by overlooking its ecumenical context, nor vice versa.

## 4. Why St. Augustine?

The discussed usage of the term "Christendom" indicates a specific (though certainly limited) independence in relation to the term "Christianity", if one would like to define the latter as a belief part, directly related to the Founder of this religion.[21] The Polish Roman-Catholic Encyclopedia, on the other hand, has a synthesis of this entry, divided into its origins, history, doctrine, contribution to culture (here, it would be closer to "Christendom") and attitude toward non-Christian religions.[22] The pursuit for the meaning of "Christendom" may in fact express a mixture of good-faith as well as political revisionism,

concerned whether there are places and times where "salvation is easier" and what it means to say that "Christians have it good or bad".

In the context of the socio-ecumenical understanding of the term in question, the work of St. Augustine should be recalled for at least several reasons. First, he seems to have wrestled with problems similar to ours: like Cardinal Suhard, he experienced a world that had come to an end, with a future that remained unknown, and whether or not the persecution of Christians would cease. Subsequently, he was a theologian of a Christianity that was not divided, or at least not divided to the degree we experience today. Thus, he could be a reference for various followers of Christ, especially since the Reformation in the classical sense was born precisely out of the Augustinian tradition. Finally, the Bishop of Hippo generates the prototype of the Church's social teaching, which, as I have tried to demonstrate in a separate study, may be taken into consideration as a vehicle for ecumenical and interreligious dialog (Krauze 2021).

St. Augustine, who, as will be expressed further in this study, strongly relativizes values, the absence of which seems to have rallied the end of Christianity in general, as follows: "Evils abound, and God hath willed that evils should abound. Would that evil men did not abound, and then evils would not abound. Bad times! troublesome times! this men are saying. Let our lives be good; and the times are good. We make our times; such as we are, such are the times. But what can we do? We cannot, it may be, convert the mass of men to a good life. But let the few who do give ear live well; let the few who live well endure the many who live ill".[23]

Now, it is impossible to omit the classic work of the author just mentioned, "The City of God", which is the first such elaborate attempt to comprehensively address the relationship between the sacred and profane from Christian positions. The circumstances of its creation in the second and third decades of the fifth century are connected with the traumatic experience of the Visigoth invasion of Rome in 410 AD.[24] The state that a century ago persecuted the young Church became a victim of the invasion of successive pagans. St. Augustine then creates a mature theology based on Platonic philosophy, a sort of prototype of Catholic social teaching, and, as could be assumed, the charter of Christendom,[25] at a time relatively close to the Constantinian breakthrough.[26] It would seem that he should reap the benefits of the new, anti-pagan order wholeheartedly. Meanwhile, in the face of phenomena that he should then vehemently deny, he remains astonishingly ambivalent. Perhaps we can better understand this attitude by following Marcus Dods, the prominent expert on St. Augustine's work in the area of English Language: "He sees that human history and human destiny are not wholly identified with the history of any earthly power—not though it be as cosmopolitan as the empire of Rome. He directs the attention of men to the fact that there is another kingdom on earth,—a city which hath foundations, whose builder and maker is God. He teaches men to take profounder views of history, and shows them how from the first the city of God, or community of God's people, has lived alongside of the kingdoms of this world and their glory, and has been silently increasing".[27]

There must have been a widespread claim in his time, to which he responds in Chapter 52, Book XVIII of the work in question: "Whether we should believe what some think, that, as the ten persecutions which are past have been fulfilled, there remains no other beyond the eleventh, which must happen in the very time of Antichrist".[28] Not only does the Doctor answer in the negative, citing examples of the difficulty of counting the number of such persecutions, but he is not at all bitter about the possibility that persecution of the Church by state institutions need not cease at all even when Christianity seems to dominate. He consistently sticks to the known facts, citing examples of contemporary persecutions, even after the Edict of Milan: under Flavius Valentinianus, by Valens in the East, or in Persia at the time. Rather, he is inclined to relativize the issue of the amount of future persecution of the Church, stating the following: "Therefore, we leave this undecided, supporting or refuting neither side of this question, but only restraining men from the audacious presumption of affirming either of them".[29]

Thus, for Augustine (1871), it is not a necessary guarantee for the City of God to continue enjoying its privileged position against possible future persecution. What is it then? Already the title of the short Chapter 20, Book XIX, brings the answer: "That the saints are in this life blessed in hope".[30] For perfect peace, which for Augustine is the supreme good of the City of God, cannot be achieved under earthly conditions, since temporality alone without eternal hope can only offer false happiness, incomparable to the goods of the soul. Also, "for that is no true wisdom which does not direct all its prudent observations, manly actions, virtuous self-restraint, and just arrangements, to that end in which God shall be all and all in a secure eternity and perfect peace".[31] It is difficult to find elements of triumphalism in this early post-Constantine narrative. At the same time, this does not mean that Augustine renounces the relationship between the City of God and the Earthly State. He states that there is some part of the City of Heaven "which sojourns on earth and lives by faith, makes use of this peace only because it must, until this mortal condition which necessitates it shall pass away"; furthermore, "as this life is common to both cities, so there is a harmony between them in regard to what belongs to it".[32]

So why the discord between the earthly and heavenly cities? According to Augustine, it was polytheism professed in the earthly state, since in the heavenly state, worship is due to one God. The matter of means of resistance against polytheism, with which he enters into irreducible conflict, is interesting in Augustine's view: "the two cities could not have common laws of religion, and that the heavenly city has been compelled in this matter to dissent, and to become obnoxious to those who think differently, and to stand the brunt of their anger and hatred and persecutions, except in so far as the minds of their enemies have been alarmed by the multitude of the Christians and quelled by the manifest protection of God accorded to them".[33] Thus, the Doctor does not demand for Christians domination in the legal order, beyond the right to profess religion, at the same time unencumbered by any specific coercive apparatus, but rather by social custom and supernatural sanction. Once again, he demonstrates an ambivalence toward any institutional domination that is astonishing to a reader accustomed to a triumphalist reading of post-Constantine church history: "This heavenly city, then, while it sojourns on earth, calls citizens out of all nations, and gathers together a society of pilgrims of all languages, not scrupling about diversities in the manners, laws, and institutions whereby earthly peace is secured and maintained, but recognizing that, however various these are, they all tend to one and the same end of earthly peace. It therefore is so far from rescinding and abolishing these diversities, that it even preserves and adapts them, so long only as no hindrance to the worship of the one supreme and true God is thus introduced".[34]

In addition to the primary reference to biblical sources, echoes of the Letter to Diogenetus, written in the blood-flow of Christians in the second century after Christ, a letter by an author who, on the one hand, does not hesitate to criticize pagan cults (and Judaism), on the other hand, without demanding privileges for Christians, on the contrary, showing their external amalgamation with the surrounding world, and compares the way they are rejected by non-Christian society to the tension that exists between soul and body.[35] As from the writings of St. Paul, Platonic dualism shines through here, although, after all, even in the post-Constantine era, mitigated by tolerance and common aspirations for peace, it is present not for political reasons, but, one might say, for metaphysical or theological ones.

## 5. Christendom and Modern Ecclesiology

In the 21st century after Christ, knowledge of what can go wrong in the relationship between the City of God and the Earth State is already quite common: wars in the name of the principle of "cuius regio eius religio" or "canonical territories", witch trials, the Stockholm syndrome of believers held hostage by far-right or left-wing political narratives and other illusions continue to recur, sometimes in a distorted mirror of the religiosity represented by some regions of the world in various stages of civilizational development. Perhaps from the depths of this moral humiliation stems the Church's modern reluctance to make clear definitions, as described most convincingly to me by Avery Dulles: "It used

to be thought, at least by many, that the Church and other realities of faith could be defined by a similar process. Thus the Church, according to Robert Bellarmine, is a specific type of human community (. . .) This clarity, however, was bought at a price. It tended to lower the Church to the same plane as other human communities (since it was put in the same general category as they) and to neglect the most important thing about the Church: the presence in it of the God who calls the members to himself, sustains them by his grace, and works through them as they carry out the mission of the Church." (Dulles 2002, p. 14).

Maybe that is why you would not find a definition of either "Christendom" or "Christianity" in the American Catholic Encyclopedia, nor the Second Vatican Council's definition of the Church. Are we doomed, or more nicely put, called to practice apophatic ecclesiology? Of course, definitions are not the only ways to teach about anything, for in the history of science, it has turned out that models and paradigms have a much greater heuristic carrying capacity,[36] as have images and models in the history of theology, beginning with parables provided by Jesus Christ himself.[37] So the "Lumen Gentium"—Dogmatic Constitution on The Church—describes her not in definitions, but in biblical images, as sheepfold, a vineyard, a building, "that Jerusalem which is above".[38] Of course, it is impossible to avoid the eschatological tension between the images of the Church presented in "Lumen Gentium" 6 and 48. In the first paragraph, we are confronted with images partially present even in the Old Testament, and in the second, with the divinely revealed predictions of the end of times, about which we can express ourselves apophatically rather than cataphatically, as St. Paul does in 1 Cor 2:9.

One level up in the new way of defining the Church is her models. As quoted by Dulles, Ian Thomas Ramsey stated that "In any scientific understanding a model is better the more prolific it is in generating deductions which are then open to experimental verification and falsification" (Ramsey 1964), which, for the former, was an encouragement to take up modeling in theology. For using models alone to describe The Infinite creates an illusion, to paraphrase the author in question, that we can enclose divine matters in the cage of our concepts, by which they themselves can become idols. Without religious experience, too, the models themselves may remain mere empty toys of a theologian who dreams he is a quantum physicist. "In so doing the theologian may not take into account the subjective element at the core of religion. The religious experience has a depth that has no correlate in our experience of the physical universe. The religious experience touches the innermost part of the person".[39]

The already mentioned biblical images cited by the last Council in describing the Church vividly resemble the "City of God"—a foretaste of the "Heavenly Jerusalem". Since "Christendom" does not easily lend itself to definitions, but neither does it exhaust itself in biblical images of the Church, nor does it appear to be a model of the Church—is another descriptive category needed in ecclesiology? Avery Dulles proposes at least seven models of the Church: Institution, Mystical Communion, Sacrament, Herald, Servant, and Community of Disciples. He proposes their evaluation as well. Should "Christendom" be placed between the models as described by Dulles, or perhaps in the still higher category proposed by the Constitution already cited? "This is the one Church of Christ (...) constituted and organized in the world as a society, subsists in the Catholic Church, which is governed by the successor of Peter and by the Bishops in communion with him, although many elements of sanctification and of truth are found outside of its visible structure. These elements, as gifts belonging to the Church of Christ, are forces impelling toward catholic unity".[40]

Would it not be to the benefit of a supra-confessional ecclesiology to bring its multidimensional imagery of the Church closer by considering the term "Christendom" as an analogate of the Kingdom of God, while "Christianity" would refer by similar analogy to "the Church of Christ"? I believe that this is one of those questions, the formulation of which we have arrived at through the linguistic and historical analyses contained in this article.[41] While remaining aware of the warning against idolatry of theological terminology (including Bellarmin's onesidedness by speaking of the Church as of "a societas"), it is worthwhile to preserve at the same time an essential element of religious experience,

recorded in St. Augustine's "City of God": that here is the Church of Christ incarnated in the reality of a given time and continent; that "Christendom" is not mere the Medieval Europe; that this is indeed the Kingdom of God, wherever it wishes to abide, wherever the Church of Christ is born.

## 6. Conclusions

The domination and state sovereignty of Christianity, customarily associated with the term "Christendom", cannot, in the light of historical data, be found at the root of its success in transforming the society of the time from polytheism to faith in the Incarnate God. Tertullian's adagium "Sanguis Martyrum Semen Christianorum" remains relevant to this day in various parts of the world and in relation to various Christian denominations. Even by reaching for modern methods of mathematical modeling we are currently unable to solve this riddle. One hundred years after Constantine, as the mature thought of St. Augustine indicates, the existence and development of the "City of God" did not depend, in essence, on the favor of the authorities of the "Earthly State", and the Doctor of the Undivided Church did not delude himself with hypothetical guarantees of the end of persecutions. God, while carrying out the construction of His Kingdom on earth, does not at all need to "roll out a red carpet" before his followers, unless this red were to signify the Wounds of Christ. Thus, on the one hand, one may be puzzled by the post-conciliar silence of Catholic publications on "Christendom", while on the other hand, the idea of the Church of Christ continuing in the Catholic Church (and, after all, the conciliar formula does not exclude its continuance in other Christian communities as well) and the return to biblical images of the Kingdom of God makes one wonder whether the term in question has already passed into oblivion, or has only assumed new garments of theological expression. After all, if "Christendom" means much more than, though traumatic for many, a caricature of Catholic religious domination or Christian cultural imperialism, then how do we address Europe's Christian heritage? Catholic theologians like Avery Dulles warn against the idolatry of defined terms, proposing instead ecclesiological models, analogous to those used in the sciences. They are distinguished from the scientific ones primarily by religious experience, which in this case cannot be replaced by anything else. As a witness to many revolutions, the term "Christendom" is more than an expression of more or less disinterested nostalgia for the "Golden Age" of medieval Europe. It rather expresses the same longing that Christians have on their lips when praying "Thy Kingdom Come"—a longing for the incarnation of Christ's Church in the realities of every Christian, regardless of denomination.

**Funding:** This research received no external funding.

**Data Availability Statement:** No new data were created or analyzed in this study. Data sharing is not applicable to this article.

**Conflicts of Interest:** The author declares no conflict of interest.

## Notes

1  Gesa Elsbeth Thiessen (Ed.), Ecumenical Ecclesiology, pp. 144–58.

2  According The Collins Online Dictionary mathematical model, the use of "Christendom" peaked in frequency ca. 1846. The old German equivalent proposed is "die Christenheit". "Das Urchristentum", on the other hand, refers to Christianity not so much spatially or organizationally, but temporally, indicating a certain time of the Church: early Christianity. https://www.collinsdictionary.com/dictionary/english-german/christendom; accessed on 12 December 2023.

3  The Oxford Dictionary of Phrase and Fable (2 ed.) 2005 Oxford University Press; https://www.oxfordreference.com/display/10.1093/oi/authority.20110803095610365; accessed on 12 December 2023.

4  https://www.merriam-webster.com/dictionary/Christendom; accessed on 12 December 2023.

5  https://www.britannica.com/event/Middle-Ages#ref908220; accessed on 12 December 2023.

6      This occurs in at least the 2003 edition, as absence between entries "Christe Sanctorum Decus Angelorum" and "Christian" p. 528. The entry "Christendom" is offered then by Francis Urquhart, The Catholic Encyclopedia, vol. 3, 1908, New York, Robert Appleton Company provided by the New Advent website: https://www.newadvent.org/cathen/03699b.htm, accessed on 12 December 2023.

7      See pages: 259, 335, 425, 434, 599, 601, etc.

8      Perhaps the Polish term "Chrystianizm" could play a role in this discussion, although it turns us back to the religious side of the meaning, as "Christian religion or doctrine". Chrystianizm, Słownik Języka Polskiego PWN, https://sjp.pwn.pl/slowniki/chrystianizm.html, accessed 12 December 2023.

9      Although the post-1945 persecution of the Church in Poland was not as horrific in scale as under Lenin and Stalin in the Soviet Union, the contrast between the earlier social acceptance of the Church and the new order was striking. One of the first-person testimonies of the persecutions are represented in writings by Primate of Poland (1948–1981), A Freedom Within: The Prison Notes of Stefan Cardinal Wyszynski (English and Polish Edition), 1984 Harcourt, San Diego. As diagnosed almost in real time by Elizabeth Valkenier (although not yet equipped with a whole image of persecutions because of the iron curtain isolation), "That is not to say that the Communists were willing to tolerate the rival claims of the Church to shape the mind a soul of the population. They merely found it wiser to pursue their goal slowly. (…) Caution was dictated by the strength and determination of the adversary: the Polish Catholic Church was a powerful institution; in prewar days it was an integral part of national life, enjoyed constitutional guarantees of its privileged position, and managed an extensive and well-knit ecclesiastical organization, together with numerous charities, schools, and a sizeable press. After the war, reinforced by a notable revival of religion and a claim the adherence of about 95 per cent of the population as a result of territorial changes, the Church strove to regain its prewar eminence in the face of radically altered political circumstances." Valkenier (1956).

10     Illustrations of the persecution of the Church in the 20th century may be found in such works as Erdozain (2017); Halík (2019); Daniel (2021). The similarity between the "creeping" communist revolution in Poland and the "creeping" secularization of the Old Continent aroused concern among Polish Roman-Catholics for the future of Europe was expressed prominently by a witness of two totalitarian systems (German Nazism and Russian communism), Karol Wojtyla, Pope John Paul II: "Everywhere, then, a renewed proclamation is needed even for those already baptized. Many Europeans today think they know what Christianity is, yet they do not really know it at all. Often they are lacking in knowledge of the most basic elements and notions of the faith (…) The great values which amply inspired European culture have been separated from the Gospel, thus losing their very soul and paving the way for any number of aberrations." Paul (2003).

11     Gianmaria Zamagni. Das "Ende des konstantinischen Zeitalters", pp. 79–80.

12     As described in Acts 2, the unusual phenomenon of glossolalia, followed by explanatory speech of Peter adds 3.000 new members to the Church instantly (Acts 2:37–42). The pattern follows in Acts 3 as well: the Holy Spirit reveals God's presence by a miracle (this time, the healing of a crippled beggar), and then Peter explains the situation calling his compatriots to conversion. However these operations cannot go unnoticed by the religious establishment of Jerusalem, as Peter persistently appeals to Jesus Christ, who by the aforementioned establishment had been considered a public enemy, which led to His execution—and in a further effect, resurrection (Acts 4).

13     "We multiply when you reap us. The blood of Christians is seed", Tertulian, Apologeticum 50:13.

14     "Peace I leave with you; my peace I give to you. I do not give to you as the world gives. Do not let your hearts be troubled, and do not let them be afraid". John 14: 27.

15     Mt 10:34-36; Lk 12:49-53. An interesting study on the attitude of the early Church to war can be found in the article: Marcin Kowalski, Holy War in Corinth: The Apocalyptic Background of Paul's Struggle against Opponents in 2 Cor 10: 3–6. Religions 2023 (14: 630).

16     Lawrence W. Reed, 2020, ISI Books, Wilmington; Was Jesus a Socialist? Why this question is being asked again, and why the answer is almost always wrong; Alexander William Salter. Jesus a Socialist? That's a Myth. The early church was egalitarian, but it wasn't committed to an economic system. 21 April 2022, Wall Street Journal https://www.wsj.com/articles/the-myth-of-jesus-socialism-acts-new-testament-christianity-catholic-protestant-marx-communism-st-paul-11650575604, accessed 12 December 2023. Seweryniak (2001).

17     As stated by Adam M. Schor, "Ultimately, a network model of Christian conversion offers suggestions on a front where standard models cannot: weighing the competition from alternative religious groups (…) [who] represented networks of their own. Each of these networks had its own set of idioms, degree of segmentation, density of relations, tolerance for external bonds, and distribution of hubs. Theoretically, it should be possible to sketch some differences in network architecture from the (scanty) evidence. This would then enable comparisons, which might reveal new explanations for Christian success, or emphasise its contingent status. While any such comparison raises the spectre of teleology, reliance on actual evidence would mitigate the danger. Network-based modelling thus offers a new (though still limited) path in the study of religious change. Realising its potential, however, requires a technique unfamiliar to most historians: computer simulation." Schor (2009, p. 497).

18     On the multilayered nature of contemporary crises and the attempt at a religiously motivated response to them: Krauze (2023a). http://www.e-transformations.com/archiwum_transformacje/2023/06/20230630202638302.pdf, accessed on 14 March 2024.

19     A popular synthesis on the modern sense of "apocalypse in history" may be found in Wojcik (1997).

[20] According to Britannica, the Church State lasted 1114 years; https://www.britannica.com/place/Papal-States, accessed 12 December 2023.

[21] According to a definition provided in the Catholic Encyclopedia by Catholic University of Lublin, Christianity is "a monotheistic religion, owing its origin to the ecclesiological activity of Jesus Christ". Romuald Łukaszyk, "Chrześcijaństwo", Encyklopedia Katolicka, vol. 3, 1995, Towarzystwo Naukowe Katolickiego Uniwersytetu Lubelskiego, Lublin, 397. As for the aforementioned American Catholic Encyclopedia, we cannot count on a strict definition accordingly. Admittedly, we can find extensive descriptions of various Christian phenomena, such as Christian philosophy and anthropology, the names of various religious congregations and church communities, Christocentrism and Christology, and even Christmas. In vain, however, as with "Christendom", look for "Christianity". Christ-Christophers. (Carson and Cerrito 2003).

[22] For more details, please see (Granat et al. 1995).

[23] Augustine (1888), Sermon 30 on the New Testament. The wider context follows: "They are the corn, they are in the floor; in the floor they can have the chaff with them, they will not have them in the barn. Let them endure what they would not, that they may come to what they would. Wherefore are we sad, and blame we God? Evils abound in the world, in order that the world may not engage our love. Great men, faithful saints were they who have despised the world with all its attractions; we are not able to despise it even disfigured as it is. The world is evil, lo, it is evil, and yet it is loved as though it were good".

[24] An example of an analysis of the historical context of the creation of "De Civitate" is the study by Leo C. Ferrari, in which the author pays particular attention to the confrontational nature of Augustine's work in relation to paganism. Reaching back to the work itself mitigates these interpretations to some extent. Ferrari (1972).

[25] "Augustine was not among those who believed that the end of the world was at hand. He left the future to God on whose providence all must depend. In the meantime he would seek to find in all merely human things the good that, as created, they must possess. If the radical division between the two cities is in the will, if love is the final determinant between their citizens, love is also the dominating quality of Augustine's book. Failure to serve the true God apart, all else he loves; all else he cherishes; all else he freely embraces. It is this calm confidence for the future and love of the created good that the Christian believer, if he is to follow the lesson of the City of God, must now in our time show." O'meara (1961, p. 113).

[26] The "City of God" has received many interpretations, such as that of Giuseppe Fidelibus, in which, in Augustine's case, the image of the City of God is the outline of the concept of the Pilgrim Church, immersed in earthly reality in order to discover God's design in it and transform it according to this design: "Augustine's work De civitate Dei warns that the epochal fall of Rome did not mark the end of civilization as such. He, precisely in defending the city of God, defended as re-formable the irreplaceable work of human thought in its original vocation to the true, in its connatural tension to the good, in its receptive openness to the beautiful. His work of thought, in an albeit incipient age of barbarism (he himself died in Hippo while the barbarians put it to the sword and fire), resists allowing itself to be led back to forms of connivance with evil and despair; the reason lies in the fact that it was born and developed in accordance with and following a much greater work: that of his civitas peregrinans to which he belonged, precisely in the act of "rebuilding destroyed cities and rebuilding devastated heritages"." Fidelibus (2012).

[27] All quotes are from Marcus Dods' translation (Augustine 1871, 1913), Volume 1 from 1913 and Volume 2 from 1871. *The works of Aurelius Augustine Bishop of Hippo. A New Translation*. vols. 1–2. The City of God. T. & T. Clark. Edinburgh. Preface, p. xi.

[28] *The City of God*, XVIII, 52, 286.

[29] *The City of God*, XVIII, 52, 288.

[30] *The City of God*, XIX, 20, 330.

[31] See note 30 above.

[32] *The City of God*, XIX, 17, 327.

[33] See note 32 above.

[34] *The City of God*, XIX, 17, 327–28.

[35] Chapter 19. of Book XIX of *The City of God* recalls this distinctive teaching already with its title: "Of the dress and habits of the Christian people". See p. 329.

[36] Ian G. Barbour, Myths, models, paradigms. A Comparative Study in Science and Religion, 1974, Harper & Row Publishers, New York-Evanston-San Francisco-London. This is one of the fundamental works helpful in carrying the dialog between science and religion in modern culture. For a theologian, Chapter 8, "The Christian Paradigm", is especially worth looking at. In this regard, reflections on modern ecumenical ecclesiology in the aforementioned article are being inspired: Cristin Colberg, Ecumenical Ecclesiology in its New Contexts, 10.

[37] Referring to the debate on the schema De Ecclesia at the first session of Vatican II, Gustave Weigel, a council peritus, observed the following in the last article published before his death: "The most significant result of the debate was the profound realization that the Church has been described, in its two thousand years, not so much by verbal definitions as in the light of images. Most of the images are, of course, strictly biblical. The theological value of the images has been stoutly affirmed by the Council. The notion that you must begin with an Aristotelian definition was simply bypassed. In its place, a biblical analysis of the significance of the images was proposed". Dulles, 16; Weigel (1963).

[38] Second Vatican Council (1964), "Dogmatic Constitution on the Church, Lumen gentium, 21 November, 1964", 6.

39   Dulles, 20–21.

40   Lumen Gentium, 8.

41   On the strength and vitality of the "Kingdom of God" category for the ecumenical movement, Pillay (2023) writes that "Fundamental to this is the realization that the kingdom lays claim not on the church but on the whole world. This turns the ecumenical movement away from self- service so that the life of the world is shifted, challenged, and transformed through the work and witness of the ecumenical movement" (from the Abstract); see also: Dicastery for Promoting Christian Unity, The Church as Community of Common Witness to the Kingdom of God. Report of the Third Phase of the International Theological Dialogue between the Catholic Church and the World Alliance of Reformed Churches (1998–2005), http://www.christianunity.va/content/unitacristiani/en/dialoghi/sezione-occidentale/alleanza-mondiale-delle-chiese-riformate/dialogo-internazionale-cattolico-riformato/documenti-di-dialogo/en.html, accessed 24 April 2024.

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
