# Peer review of "Complexity and Timeliness of the Term “Christendom” for Ecumenical Ecclesiology"

_religions, doi:10.3390/rel15050592_

Round 1

Reviewer 1 Report

Comments and Suggestions for Authors

Reference should be made to  Gianmaria Zamagni, Das „Ende des konstantinischen Zeitalters“ und die Modelle aus der Geschichte für eine „neue Christenheit“. Eine religionsgeschichtliche Untersuchung, Freiburg i. Br. 2018.

This said, there should be mentioned the debate in the 1930s to 1950s, and also the later theological discussion, see e.g.,  Karl Gabriel, Christentum zwischen Tradition und Postmoderne, Freiburg 1992.

An interesting and after all convincing point is the sensibility for the relationship between politics and religion in general (ch. 3). As to the concrete way to deal with the possible use of "Christendom", the choice to be very close to Augustin (ch. 4) is interesting, but does not completely convince. It should be better explained why Augustin is introduced and what exactly one can learn from him (and what not).

The link between Augustin and the ecclesiology of Vatican II does not entirely convince. For LG speaks in images, but the structure is trinitarian (DV 2-4) and the focus is "mysterium" (chapter 1) and "people of God" (chapter 2). The "City of God" in an augustinian sense is not at the center. Also: is LG 6 more important than LG 48? There lacks a coherent hermeneutic of Vatican II. The reference to Dulles does not replace this, for the conclusions taken with reference to Dulles can be questioned. 

For the sacramentality in LG and the eschatology of LG (togetherness of, but also distinction between earthly and heavenly Church) I wonder if one can so easily link so close "Christendom" to "Kingdom of God" (p. 10). At least, this seems to go much too fast.  This critical point must be maintained even though the term could indeed, as it is said in the conclusion, help further in ecumenical discussions. But then, Christendom is not the Kingdom of God, but a way to describe as the social dimension of the Church (nevertheless: attention not to repeat Bellarmin's onesidedness by speaking of the Church as a societas)

Author Response

Dear Reviewer,

I would like to express my gratitude for the help you are giving me in better preparing this publication. I bought Gianmario Zamagni's book and have no regrets. Without its inclusion, and likewise concerning the complementary reading of Karl Gabriel, my study would have been blemished. In the new version of the article, I explicitly refer to the proposed corrections. I have renamed the title of Chapter III "Why St. Augustine?" and there I give some seemingly obvious motives for his importance to ecumenical ecclesiology. But it should have been written openly. Thank you for your encouragement to better analyze Lumen Gentium. You have also prompted me to correct a nasty mistake: of course, it is heresy to write that "Christendom" is equivalent to the Kingdom of God! Rather, it is Its analogate. I agree, so much so that I borrowed the given remark on Bellarmine. In the submitted working file, the corrections were made in yellow.

With my best regards,

the author

PS. Updates for a parallel review are marked in green.

Reviewer 2 Report

Comments and Suggestions for Authors

The subject addressed by this paper is a really interesting one, and I would want to encourage the author to keep exploring it. Awareness of different languages is effectively deployed, and the use of Augustine a helpful contribution to a contemporary debate. But this is a huge subject, and I didn't see enough engagement with recent or current contributions to it (I also missed any acknowledgement of Markus' landmark study on City of God, which would be highly relevant to the subject).  In terms of ecumenical ecclesiology, it would be important to recognise the diverse stances regarding 'Christendom' (understood as a particular category of relation between a church and the society to which it belongs, which characterized some parts of Europe in centuries past) of different participants in contemporary ecumenism, not least because of their different histories. The 'national' churches of the Reformation, for instance, represented a rather different kind of Christendom from medieval European Catholicism, while other churches have developed in conscious opposition to Christendom (e.g. Baptists) or - more recently - in something like indifference.

Perhaps the fundamental theological question here is whether Christendom in this sense is at all desirable for the church, and hence whether its passing is to be mourned, rejoiced over, or treated with detachment (the Augustinian response?). While the final sentences of the abstract indicate at least an ambivalent answer here, the article as it unfolds seems more consistently negative in its assessment, with the concluding sentence only finding positive meaning in a sense of the word wholly removed from its historical context.

Comments on the Quality of English Language

The quality of English language varies significantly in this article, with some of the text more clearly and accurately written than others.

Author Response

Dear Reviewer,
thank you for taking the time to familiarize yourself with my article and for the many words of encouragement for further work found in the submitted Review. I was driven primarily by curiosity as to what differences in meaning can be found between "Christianity" and "Christendom," which are virtually invisible in my native language, but which can be pointed out in English or German. So I focused on a linguistic and historical study, the natural, though not obvious at first glance, consequence of which was to find ecumenical meanings. If I understood correctly that the point about Marcus is, of course, to take into account the translation work of Marcus Dods, then I agree, and thank you for suggesting the Edinburgh edition as a reference. I have marked the corrections resulting from this in green. As for the article as a whole, perhaps the corrections suggested by the other reviewers (in yellow) will bring a little more clarity to the proposed narrative. For me, this is only a starting point, though sincerely earned, from my Roman Catholic perspective toward a broad understanding of "Christendom" in relation to the "Kingdom of God" as an ecumenical reality. I hope that I have at least partially addressed your helpful remarks.
With best regards,

the author

Reviewer 3 Report

Comments and Suggestions for Authors

 In the five chapters of the manuscript the author achieves to present the issue of complexity of the term Christendom and the central place that it has in history. Additionally, the author demonstrates a good knowledge of paragraph development, grammar, punctuation and composition skills, while he/she focuses on the main subject avoiding any sort of repetition. Due to this methodology the writer supported balance in the work, free from bias, offering to the reader the opportunity to have an objective perspective of the theme.

Taking into consideration that it is an issue with manifold parameters therefore, I make a recommendation of accept and publish the manuscript, because in my opinion it is worthwhile and it could contribute to the field of theology. It goes without saying that these comments express only my personal view. 

Author Response

Dear Reviewer,
Thank you for making the effort to review my article. The enclosed review is a great encouragement for me to sail into the wide waters of international publications. I hope we will meet someday because the world of religious ideas and beliefs is a very vast and interdisciplinary space.
With best regards and gratitude,

the author

PS. As a result of comments provided by other reviewers, I had the opportunity to supplement the study with some new content, which I have highlighted in yellow. I hope that they will also meet with your positive reception.

PS. Updates for a parallel review are marked in green.

Round 2

Reviewer 1 Report

Comments and Suggestions for Authors

The additions are a good improvement. The article sufficiently mentions that the term "Christendom" can be misleading, but now unfolds its potential in a sufficiently prudent manner. The article can therefore be seen as an original contribution to the ecumenical debate.